# *Pseudomonas simiae* WCS417 Strain Enhances Tomato (*Solanum lycopersicum* L.) Plant Growth Under Alkaline Conditions

**DOI:** 10.3390/plants14020264

**Published:** 2025-01-18

**Authors:** Miguel A. Aparicio, Francisco J. Ruiz-Castilla, José Ramos, Francisco J. Romera, Carlos Lucena

**Affiliations:** 1Departamento de Agronomía, Edificio Celestino Mutis (C-4), Campus de Excelencia Internacional Agroalimentario de Rabanales (ceiA3), Universidad de Córdoba, 14071 Córdoba, Spain; bio.angel.zoo@gmail.com (M.A.A.); ag1roruf@uco.es (F.J.R.); 2Departamento de Química Agrícola, Edafología y Microbiología, Edificio Severo Ochoa (C-6), Campus de Excelencia Internacional Agroalimentario de Rabanales (ceiA3), Universidad de Córdoba, 14071 Córdoba, Spain; fjruizcastilla@hotmail.com (F.J.R.-C.); mi1raruj@uco.es (J.R.)

**Keywords:** *Pseudomonas simiae*, tomato, alkaline conditions, inoculation, bicarbonate, hydroponic system, solid substrate

## Abstract

Iron (Fe) deficiency is among the most important agronomical concerns under alkaline conditions. Bicarbonate is considered an important factor causing Fe deficiency in dicot plants, mainly on calcareous soils. Current production systems are based on the use of high-yielding varieties and the application of large quantities of agrochemicals, which can cause major environmental problems. The use of beneficial rhizosphere microorganisms is considered a relevant sustainable alternative to synthetic fertilizers. The main purpose of this work has been to analyze the impact of the inoculation of tomato (*Solanum lycopersicum* L.) seedlings with the WCS417 strain of *Pseudomonas simiae*, in the presence or absence of bicarbonate, on plant growth and other physiological parameters. To conduct this research, three different inoculation methods were implemented: root immersion, foliar application, and substrate inoculation by irrigation. The results obtained show the ability of the *P. simiae* WCS417 strain to induce medium acidification in the presence of bicarbonate to increase the SPAD index and to improve the growth and development of the tomato plants in calcareous conditions provoked by the presence of bicarbonate, which indicates that this bacteria strain could have a great potential as an Fe biofertilizer.

## 1. Introduction

From a microbiological perspective, soil is probably the most complex of all natural environments due to the size of its microbial community and its diversity. It is estimated that a gram of cultivated or pasture soil contains about 2 × 10^9^ prokaryotic cells [1].

The portion of the soil in contact with plant roots is called the rhizosphere, ranging from 2 to 80 mm from the root [2]. The rhizosphere harbors a rich microbial community of up to 10^10^ bacteria per gram of soil [3,4] and encompasses a great diversity of taxa [5,6]. The majority of available information regarding the plant–microbe relationship is based on the pathogenic effect of microorganisms on plants [7]. However, despite numerous studies focused on phytopathology, most microorganisms do not cause harm to plants; on the contrary, many exert beneficial effects [8,9] by promoting plant growth and improving health through direct and indirect mechanisms [10,11].

Beneficial interactions between plants and microorganisms are symbiotic interactions where costs and benefits are shared by both [12,13]. Mutualistic symbioses correspond to the intimate, and mostly obligatory, interactions between microorganisms and a strict range of compatible host plants. They generally require the formation of structures specifically dedicated to this interaction (e.g., nodules during symbiosis between rhizobia and legumes, arbuscular mycorrhizas) [14,15,16]. These interactions also involve soil bacteria capable of colonizing the root system’s surface (sometimes even internal root tissues) and stimulate plant growth and health, known as plant growth-promoting rhizobacteria (PGPR) [17]. The colonization of roots by PGPR is heterogeneous along the root system, and their competitiveness during this process is an essential condition for the promotion of plant growth [18,19,20]. In contrast to mutualistic interactions, PGPR interact with a wide range of plant species and encompass a broad taxonomic diversity, especially within the Firmicutes and Proteobacteria phyla [20,21].

The mechanisms by which development and growth promotion occur can be diverse. For example, they can occur through the solubilization of specific nutrients, the production of certain compounds that directly promote growth (exudates), or the stimulation of the production of phytohormones [22]. Regarding the increase in nutrient supply through solubilization, the following two main types of bacterial activities need to be considered: (i) the production of organic acids, and (ii) the release of H^+^ to the outer surface inducing the H^+^-ATPase action. It is well known that many PGPR, such as species from the genera *Pseudomonas*, *Bacillus*, or *Rhizobium*, can solubilize insoluble forms of phosphate [11] by acidifying the external medium through the secretion of low molecular weight organic molecules that chelate cations bound to phosphate [23], or by producing phosphatases and phytases that hydrolyze phosphate-rich organic molecules. Another example is the activation of the plants machinery necessary for Fe acquisition, induced by the production of volatile organic compounds emitted by *Bacillus subtilis* GB03. These compounds act at two levels; firstly, they acidify the medium and subsequently induce the expression of genes encoding various molecules involved in Fe assimilation, including a Fe^3+^ chelate reductase enzyme, a Fe^2+^ transporter, and a transcription factor that induces the expression of these two genes [24,25]. Another important aspect of soil microbiology is the action of well-known Fe-bacteria, particularly, schizomycetes capable of promoting oxidative processes at the expense of Fe compounds. In calcareous soils where the bicarbonate fraction is in equilibrium with the carbonate fraction, ferrous carbonate is oxidized to insoluble ferric hydroxide with the release of CO_2_ by the chemoautotrophic Fe-bacteria in the soil [26].

On the other hand, many bacteria fix the atmospheric nitrogen that becomes available to the plant. There is evidence supporting the involvement of PGPR in the N_2_ supply to various plants, especially sugarcane [27,28,29]. However, the impact of N_2_ fixation by PGPR is still debated, and it is not entirely clear whether the nitrogen fixed by these bacteria substantially contributes to plant growth [30,31,32]. Finally, some non-nitrogen-fixing rhizobacteria promote plant growth, demonstrating that there must be other mechanisms, independent of nitrogen supply, by which these bacteria promote plant growth. For example, *Phyllobacterium brassicacearum* STM196 is unlikely to fix N_2_, yet it has been shown to promote the growth of rapeseed and Arabidopsis thaliana plants [33,34,35].

Regarding exudate production, microorganisms can act on the plant by modulating the composition of its exudates, contributing to the recruitment of beneficial ones and the suppression/inhibition of non-beneficial and pathogenic ones to avoid infections and the proliferation of non-beneficial microbial communities [24]. These exudates also interact with organic and inorganic molecules in the soil, regulating their availability in the soil and their absorption by the plant [36]. The growth and composition of the root microbial environment are largely determined by the composition of radical exudates, as many of these compounds attract bacteria, especially those that can metabolize some of their components and proliferate in that environment [20,37,38,39]. Also, concerning the stimulation of phytohormone production, certain plant responses are modulated by ethylene. Ethylene produced due to stress factors (extreme temperature increase, presence of heavy metals, lack of water, nutritional deficiencies, and the presence of microorganisms) can lead to responses that may enhance plant survival in adverse conditions. Various microbial strains have been shown to induce ACC synthase, which catalyzes the production of ACC, a precursor of ethylene, from S-adenosylmethionine [40]. It is interesting to note the effect that some microorganisms have on certain plant species. For example, the inoculation of *Pachycereus pringlei* seedlings with the bacterium *Azospirillum brasiliense* can not only promote plant growth but also reduce the pH of the rhizosphere [41].

Although the experiments have been conducted in hydroponic solution, we cannot overlook the presence of humic and fulvic acids in the soil. Considering that the study focuses on calcareous substrates, it is appropriate to highlight the following: (a) Humic acids have chelating properties, binding to nutrients in the soil and facilitating their uptake. (b) There exists high cation exchange capacity (CEC). (c) Humic acids are water-soluble at alkaline pH levels. Their highest concentration is found in calcareous or neutral soils [42,43].

Bicarbonate is a key factor contributing to iron (Fe) chlorosis in dicot plants, particularly in calcareous soils [44,45,46]. However, its exact mechanism of action remains unclear. Traditionally, the primary effect of bicarbonate on Fe nutrition has been attributed to its pH-buffering capacity. By maintaining a high pH level (7.5–8.0) in the surrounding medium, bicarbonate can reduce Fe solubility and inhibit ferric reductase activity, which operates optimally at around pH 5.0 [47,48]. While this pH-buffering effect is significant, other studies indicate that bicarbonate may impair Fe nutrition by inhibiting ferric reductase activity and limiting other Fe-acquisition mechanisms [49,50,51,52,53].

Bicarbonate’s inhibition of Fe responses may also be linked to its suppressive effect on transcription factors such as *SlFER* and *AtFIT*, and subsequently, on Fe-acquisition genes like *FRO*, *IRT*, and *CsHA1* [51]. Conversely, other studies have observed that bicarbonate can stimulate ferric reductase activity and the expression of Fe-acquisition genes [49,51,54]. These seemingly contradictory findings could be attributed to variations in experimental conditions, such as differences in bicarbonate and Fe concentrations, the treatment duration, and plant species. For instance, Lucena et al. [51] applied bicarbonate and 0 μM Fe to *Arabidopsis* plants for two days, whereas Msilini et al. [55] used bicarbonate and 5 μM Fe over a 30-day period. Generally, bicarbonate appears to inhibit ferric reductase activity and other Fe responses more significantly at higher concentrations and in the absence of Fe [51]. This suggests that bicarbonate’s influence on Fe nutrition is more complex than its pH-buffering capacity alone would indicate.

In addition to bicarbonate, several environmental factors can trigger or worsen Fe chlorosis in soils, including high moisture, poor drainage, inadequate aeration, and soil compaction—each associated with hypoxic stress, which can exacerbate Fe chlorosis [44,45,46,56,57,58]. Most studies link hypoxia’s negative impact on Fe nutrition to its role in increasing soil bicarbonate levels. García et al. [52] provided initial evidence that hypoxia may directly inhibit Fe-acquisition gene expression. Under these conditions, elevated bicarbonate levels may further compromise Fe nutrition [45,46,56,58,59]. In waterlogged or compacted soils, water saturation fills soil pores, raising the partial pressure of CO_2_ in the soil air and, consequently, the bicarbonate concentration [44,45,46,57].

The WCS417 strain is known to be a plant growth promoter. It has been determined that in *Arabidopsis*, it can favor both the increase in the size of the aboveground part in in vitro and substrate cultures, as well as in chlorophyll levels. Similarly, it also affects root structure, promoting greater development, resulting in the formation of more lateral roots [60]. Other strains belonging to the *Pseudomonas* genus have been shown to have a growth-promoting effect on *A. thaliana* seedlings due to the production of pyoverdine, a siderophore synthesized by the bacterium under iron-deficient conditions [61]. Under general nutrient deficiency in the soil and high pressure or a CO_2_ environment, the WCS417 strain is capable of both promoting plant growth expressed as fresh weight and increasing leaf surface [62]. It is known that auxins produced by certain microorganisms are molecules attributed to this growth-promoting effect [63]. In recent years, it has been discovered that this growth-promoting effect is dependent on auxin synthesis by the WCS417 strain [63,64]. Inoculation with the WCS417 strain or exposure to volatiles produced by it can induce the transcription of other genes involved in iron deficiency in *Arabidopsis* seedlings, such as the *IRT1* and *FRO2* genes [65,66].

The objective of this article has been to determine the effect of inoculation with the WCS417 strain of *P. simiae* on the physiological response mechanisms to iron deficiency in tomato seedlings and to study the possible effect of this bacterial strain on promoting the growth of tomato plants in alkaline pH.

## 2. Materials and Methods

### 2.1. Bacteria Strain, Plant Variety, Growth Conditions

*Pseudomonas simiae* WCS417 was cultured at 27 °C in King’s B medium [67] (20 g/L, 1.5 g/L K_2_HPO_4_, 1.5 mM MgSO_4_, 15 mL/L glycerol, pH 7.2 ± 0.2), supplemented with Rifampicin (50 μg/mL). The cells were preserved in glycerol at −80 °C. Regarding the tomato seeds used, this study employed tomato seeds of the ‘Tres Cantos’ variety.

### 2.2. Seed Germination

Seeds were subjected to surface sterilization using 20% sodium hypochlorite solution for 1 min, followed by rinsing with distilled water. Subsequently, the seeds were placed on a moist perlite substrate in a tray. A 5 mM CaCl_2_ solution (20 mL) was used to facilitate germination. The seeds remained under growth chamber conditions. Around 20–25 days after planting in a tray with perlite, the seedlings had already developed cotyledons, and the first two true leaves were beginning to emerge. Carefully, they were individually removed from the perlite, their roots were washed with ample distilled water, and they transferred to the hydroponic cultivation system [68]. Each individual plant was inserted in plastic lids and held in the holes of a thin polyurethane raft floating on an aerated nutrient solution containing 2 mM Ca(NO_3_), 0.75 mM K_2_SO_4_, 0.65 mM MgSO_4_, 0.5 mM KH_2_PO_4_, 50 µM KCl, 10 µM H_3_BO_3_, 1 µM MnSO_4_, 0.5 µM CuSO_4_, 0.5 µM ZnSO_4_, 0.05 µM (NH_4_)_6_Mo_7_O_24_, and 20 µM Fe-EDDHA.

This hydroponic system was maintained in the growth chamber at 22 °C day/20 °C night temperatures with a relative humidity of 70% and a 14 h photoperiod at an irradiance of 300 µmol m^−2^ s^−1^.

### 2.3. Cultivation of Bacteria and Inoculum Preparation

The *P. simiae* WCS417 inoculum was obtained from a stock preserved in glycerol at −80 °C. They were cultured on KB agar plates (King’s medium B) [67] supplemented with 50 μg mL^−1^ of rifampicin at 27 °C for 24 h. Subsequently, the cells were harvested using 10 mM MgSO_4_ after being washed twice by centrifugation for 5 min at 4500× *g*. Finally, the cells were resuspended in 10 mM MgSO_4_, and the optical density (OD) at 600 nm was determined before inoculation.

### 2.4. Experimental Conditions

Plants aged 20–25 days were used for the experiments. The plants were selected to ensure similar size and growth, aiming to minimize potential variability among seedlings.

Treatments were designed with a concentration of 40 μM Fe-EDDHA (until that moment, the plants had been growing with a lower Fe concentration of 20 μM Fe-EDDHA), and each plant was housed in an individual 0.5 L container, as detailed earlier, connected to an aeration system to prevent anoxic conditions. To simulate the alkaline conditions of calcareous soils, 40 mM NaHCO_3_ was added to the nutrition solution, since bicarbonate is one of the most important factors inducing Fe chlorosis [51]. Three different types of inoculation were carried out as follows: (1) inoculation in nutrient solution; (2) inoculation through foliar spraying; and (3) inoculation through substrate irrigation. Based on the various variables considered in this experimental design, the following treatments were applied: Control, Inoculation in solution, Foliar inoculation, Control + NaHCO_3_, Inoculation in solution + NaHCO_3_, and Foliar inoculation + NaHCO_3_. For the substrate experiments conducted in 0.7 L capacity pots, the following two types of different solid substrates were used: (1) perlite and (2) black peat. For each substrate type, a control treatment and an inoculated treatment were set up. Twelve tomato plants were used for each treatment.

Regarding the inoculation, in all cases, the plants were inoculated with a concentration of 10^7^ CFU mL^−1^ of bacteria. For foliar inoculation, the aerial part of the seedlings was sprayed with a bacterial suspension containing 10^7^ CFU mL^−1^ using a sprayer. In substrate-based trials, the seedlings were irrigated with 100 mL of a bacterial suspension containing 10^7^ CFU mL^−1^.

### 2.5. Physiological Determinations

#### 2.5.1. Growth Promotion

Periodic measurements of the height of all treatment plants were taken. At the time of plant harvest, the fresh and dry weight of both the aboveground and root parts were determined.

#### 2.5.2. SPAD

The level of chlorosis in the plants was determined through SPAD readings. A portable chlorophyll meter, Minolta SPAD-502 (Konica Minolta, Tokyo [Japan]), was employed. Three readings per leaf were taken, and the arithmetic mean of these readings was calculated.

#### 2.5.3. pH Determination

In the experiments conducted in hydroponic cultivation, the pH of the nutrient solution was monitored using a portable pH meter. Different pH meters were used to prevent potential contamination. After the determinations, the pH meter electrode was washed with a bactericide.

#### 2.5.4. Bacterial Colonization

To measure the colonization and survival of bacteria under the study conditions, samples of both root and nutrient solutions were taken at two intervals, one at 26 days and the other at 37 days after inoculation. These samples were inoculated onto rifampicin-supplemented KB plates at 27 °C for one day. Subsequently, the number of colony-forming units (CFU g^−1^ for root samples and CFU mL^−1^ for nutrient solution samples) was calculated.

### 2.6. Statistical Analysis

The statistical analysis and graphs were performed using GraphPad Prism 8. To compare data from the treatments, Student’s *t*-test was used for parametric data and the Mann–Whitney test for non-parametric data. For multiple comparisons, the ANOVA test was used for parametric data and the Kruskal–Wallis test for non-parametric data. The significance level was determined by asterisks, with * *p* < 0.05, ** *p* < 0.01, or *** *p* < 0.001 indicating the presence of significant differences between the treatments.

## 3. Results

### 3.1. Monitoring the Growth of Tomato Plants and the External pH After Inoculation in the Nutrient Solution

Inoculation with the WCS417 strain in tomato plants grown in a hydroponic system supplemented with 40 μM Fe-EDDHA showed no visible effects on growth, as determined by plant length (Figure 1B). Monitoring the nutrient solution pH revealed significant differences only at 21 days, although the trend suggested that plants inoculated with the bacteria tended to acidify the medium (Figure 1D). On the other hand, in the presence of 40 mM NaHCO_3_ (sodium bicarbonate) with a strong alkalizing effect on the medium, the inoculated plants exhibited greater growth than the control plants from day 15 onwards. The control plants ceased to grow (Figure 1A). After 37 days, at the final harvest, 4 out of 6 of the remaining control plants had dried up, while the inoculated treatment plants remained alive (Figure 1E). Regarding the pH, the WCS417 strain triggered the acidification response in plants starting from the fifth day after inoculation (Figure 1C).

Additionally, two harvests were conducted, each consisting of six randomly selected plants from each treatment, one at 26 days and another at 37 days. Both the aboveground and root parts of the plants were weighed. In the first harvest, there were no significant differences in growth between plants from the inoculated and control treatments. In the second harvest, the hydroponically grown inoculated plants in the absence of NaHCO_3_ exhibited a significantly greater root weight than the control treatment plants. There were no significant differences for the aboveground part (Figure 2A). On the other hand, in the presence of 40 mM NaHCO_3_, the inoculated plants showed a significantly higher weight for both the root and aboveground parts compared to the control treatment plants (Figure 2B).

When counting bacteria during the first harvest, it was determined that both in the nutrient solution and in the roots, the bacterial count was much higher in the absence of NaHCO_3_ (Figure 3A,C). On the other hand, in the second harvest, the number of CFUs per gram or milliliter clearly decreased compared to the first harvest. In the presence of NaHCO_3_, the bacteria had completely disappeared in the nutrient solution. In the absence of NaHCO_3_, there were some samples where bacterial presence still persisted (Figure 3D). In the roots, although the quantity of bacteria decreased significantly compared to the first harvest, the survival was higher compared to the nutrient solution. Survival was also higher in the absence of NaHCO_3_ (Figure 3B).

### 3.2. Monitoring the Growth of Tomato Plants and the External pH After Foliar Inoculation

Tomato plants cultivated in the presence of NaHCO_3_ experienced a growth-promoting effect through foliar inoculation with the WCS417 strain. In contrast, plants in the control treatment did not exhibit any growth (Figure 4A). Foliar inoculation in this same treatment favored the root’s rhizosphere acidification response, as shown in Figure 4C. Chlorophyll levels in the control treatment gradually declined, becoming significant only at the end of the trial on day 15 (Figure 4E). There is nothing noteworthy regarding plants grown in the absence of NaHCO_3_, as there were no significant differences between the control and the inoculated treatment plants (Figure 4B,D,F).

### 3.3. Effects of Inoculation with the WCS417 Strain on Tomato Plants Cultivated in Pots with Solid Substrate (Peat or Perlite)

Inoculation with the WCS417 strain only had significant effects on the growth of tomato plants cultivated in black peat (Figure 5A). Regarding chlorophyll levels, no significant differences were detected between the control and inoculated treatments, either in black peat or perlite (Figure 5C,D). In perlite, there were also no significant differences between the control and inoculated treatments in terms of growth over time (Figure 5B).

## 4. Discussion

The WCS417 strain of *P. simiae* is one of the best-characterized PGPR. It can colonize the external surface of the roots of plants such as *Arabidopsis thaliana*, and it also has the ability to colonize endophytically tomato plants [63]. There is evidence suggesting that the growth-promoting effect and changes in root architecture are due to the production of auxins by this strain [69]. Similarly, it is credited for the promotion of growth along with other mechanisms, such as inhibiting the growth of other microorganisms [70]. The inoculation of dicotyledonous plant roots with the WCS417 strain has been shown to trigger the induction of ISR in distal tissues, providing protection against a wide range of pathogens and herbivores [63]. It is known that the WCS417 strain can induce the *MYB72* gene, encoding the homonymous transcription factor involved in both ISR and the response to Fe deficiency [70]. In the same line, it has been demonstrated that volatiles produced by the WCS417 strain can induce the expression of genes related to Fe deficiency in *A. thaliana* seedlings [66].

The results obtained in this study demonstrate the growth-promoting effect of the WCS417 strain on tomato plants grown in a hydroponic system subjected to nutritional stress due to the presence of bicarbonate (Figure 1A, Figure 2B and Figure 4A). Other authors have already highlighted this growth-promoting effect [71]. Similarly, inoculating *Vigna radiata* plants with the GRP3A strain of *Pseudomonas* clearly produced a growth-promoting effect under both Fe sufficiency and deficiency conditions. Additionally, this strain is capable of producing siderophores under Fe deficiency conditions, aiding the plant in overcoming conditions of Fe deficiency [72]. Similarly, Ali et al. [73] have demonstrated that inoculating wheat plants with the AKMP7 strain of *Pseudomonas putida* promoted both root and shoot growth (measured by length and dry weight). Similarly to what is shown in this chapter with the inoculation of tomato plants in the presence of bicarbonate, Sandhya et al. [74] demonstrated that five different *Pseudomonas* strains were able to induce growth promotion in maize plants cultivated under water stress conditions, demonstrating the agronomic potential of members of this bacterial genus under different environmental stress conditions.

In relation to the effect of the WCS417 strain of *P. simiae* on foliar application, it is worth highlighting that our results are still preliminary. Our research group’s subsequent investigations will focus on elucidating the mode of action of the strain of *P. simiae.* We hypothesize that foliar application triggers a series of signaling mechanisms that travel from the leaf to the root, activating the mechanisms responsible for responding to nutrient deficiencies [9,25,75]. The activation of that signaling route could be the response leading to growth promotion, medium acidification, or the SPAD value incrementation in inoculated plants. Previous studies by our group have shown that hormones such as ethylene or nitric oxide are involved in these signaling systems [9,25,75].

Regarding alkaline pH levels, it is known that growth-promoting microorganisms, especially alkali-tolerant ones, can promote plant resistance to these conditions by producing IAA (auxins) and ACC, and by allowing an increase in internal potassium levels. In this way, the plant’s relative humidity and ionic homeostasis are maintained [76]. Ipek et al. [77] used five different strains with a growth-promoting effect on strawberry plants cultivated in calcareous soil, resulting in a highly significant growth-promoting effect across all strains. However, inoculation with the WCS417 strain only had relevant effects on the growth of tomato plants grown in peat (Figure 5A). Regarding chlorophyll levels, no significant differences were detected between the control and inoculated treatment in either peat or perlite (Figure 5C,D). In perlite, no significant differences were detected between the control and inoculated treatment in terms of growth over time (Figure 5B). Black peat and perlite are different solid substrates with distinct textures and compositions. In the preliminary trials that shaped this paper, a viability study of the bacterium in both substrates was not conducted. From our perspective, inoculation through irrigation was more successful in peat than in perlite because it was more capable of remaining in the medium. The inoculum may have been washed away with subsequent irrigations in perlite but not in peat. For future experiments, we plan to conduct viability analyses in these and other substrates (cultivated soils) to confirm whether or not the bacterium is suitable for use.

It is interesting to note that, at 26 and 37 days after the application of the treatments, tomato roots still showed the presence of CFU g^−1^ both in the presence and absence of bicarbonate (Figure 3A,B). However, bicarbonate reduced the survival of the bacterium in both the nutrient solution and the roots. In both cases, there was a decrease in the number of CFU g^−1^ compared to the roots and solution of tomato plants cultivated in the absence of NaCO_3_ (Figure 3). More specifically, NaCO_3_ caused the disappearance of the bacterium toward the end of the experiment in the nutrient solution. However, the WCS417 strain still existed in the plant roots (Figure 3B). Similarly, Carrillo et al. [41] also detected root colonization by the GB03 strain of *B. subtilis* at the level of 10^6^ CFU mg^−1^. The number of CFU mg^−1^ in this case was influenced by the nitrogen source used in the nutrient solution. Ahmad et al. [78] used several strains to test the root colonization of cotton plants after one week. They determined that two strains, one of *B. subtilis* and another of *Paenibacillus* sp., showed high efficiency in colonization, almost reaching the order of 10^6^. These also had a strong growth-promoting effect on the plant. Some years ago, Lucena et al. [51] demonstrated that bicarbonate could cause Fe deficiency by inhibiting the expression of several Fe acquisition genes in dicot plants. Authors underlined the difference between this bicarbonate effect (inhibition of expression of Fe acquisition genes) and other effects due to its pH buffer capacity. In calcareous soils, the ability of plants to induce the ferric reductase, the iron transporter, and the H^+^-ATPase genes allows them to locally acidify (by the subapical regions of the roots) the rhizosphere and to mobilize Fe from soil particles. However, if the expression of the genes is blocked by a high concentration of bicarbonate in the soil (as found by Lucena et al. [51]), this capacity does not exist, and plants are not able to remobilize Fe from the soil. In conclusion, the *P. simiae* WCS417 strain induces an acidification response in tomato plants cultured in the presence of 40 mM NaCO_3_. This acidification was observed both when the bacteria were inoculated by foliar spraying and by root immersion. In addition, our results demonstrate the growth-promoting effect of the WCS417 strain in tomato plants subjected to nutritional stress due to the presence of bicarbonate.

## 5. Conclusions

The results presented in this work show the ability of the *P. simiae* WCS417 strain to induce medium acidification in the presence of bicarbonate, increasing the SPAD index and improving the growth and development of tomato plants in calcareous conditions provoked by bicarbonate presence. This indicates that this bacteria strain could have a great potential as an Fe biofertilizer. However, further research is necessary to fully understand its mechanisms of action and to optimize its application for agricultural purposes. Continued investigation and experimentation is crucial for harnessing the full potential of the *P. simiae* WCS417 strain as a biofertilizer that enhances Fe uptake and improves crop productivity in tomato, and potentially other agricultural systems and species. The potential of the *P. simiae* WCS417 strain to be used as an Fe biofertilizer opens up new possibilities for its application in more sustainable and environmentally friendly agriculture.

## Figures and Tables

**Figure 1 plants-14-00264-f001:**
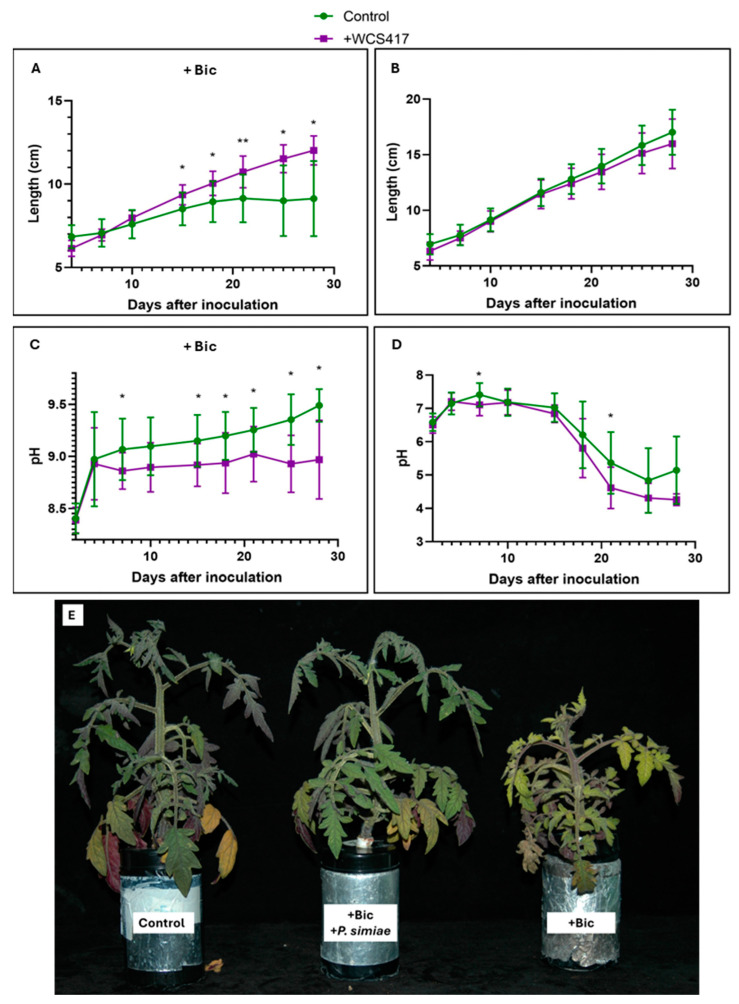
Effect of inoculation with the WCS417 strain on the growth and acidification of tomato plants grown in a hydroponic system. Once the nutrient solution was inoculated with 10^7^ CFU mL^−1^ bacterial suspension of the WCS417 strain, seedlings were transferred to individual containers. Shoot growth and the pH of the nutrient solution were monitored over time. (**A**) Growth evolution of tomato plants over 28 days cultured in the presence of 40 mM NaHCO_3_. (**B**) Growth evolution of tomato plants over 28 days cultured in the absence of NaHCO_3_. (**C**) pH evolution in the nutrient solution over 28 days of tomato plants grown in the presence of 40 mM NaHCO_3_. (**D**) pH evolution in the nutrient solution over 28 days of tomato plants grown in the absence of NaHCO_3_. (**E**) Comparison between control, inoculated treatments with *P. simiae* in the presence of NaHCO_3,_ and non-inoculated treatments of tomato plants grown in the presence of NaHCO_3_ and after 37 days. * *p* < 0.05 and ** *p* < 0.01 indicate significant differences between inoculated and control treatments. Values are the means ± S.E. of 12 replicates.

**Figure 2 plants-14-00264-f002:**
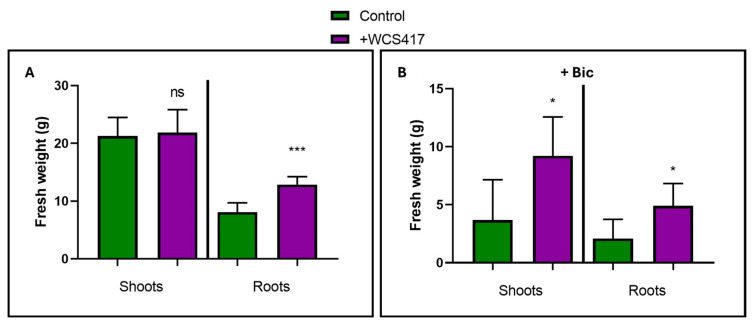
Growth promotion of shoots and roots of tomato plants inoculated with the WCS417 strain after 37 days of cultivation in a hydroponic system. Shoots were excised from roots and weighed separately. (**A**) Tomato plants grown in the absence of NaHCO_3_. (**B**) Tomato plants grown in the presence of 40 mM NaHCO_3_. * *p* <0.05 and *** *p* < 0.001 indicate significant differences between inoculated and control treatments, ns (no significant differences). Values are the means ± S.E. of 12 replicates.

**Figure 3 plants-14-00264-f003:**
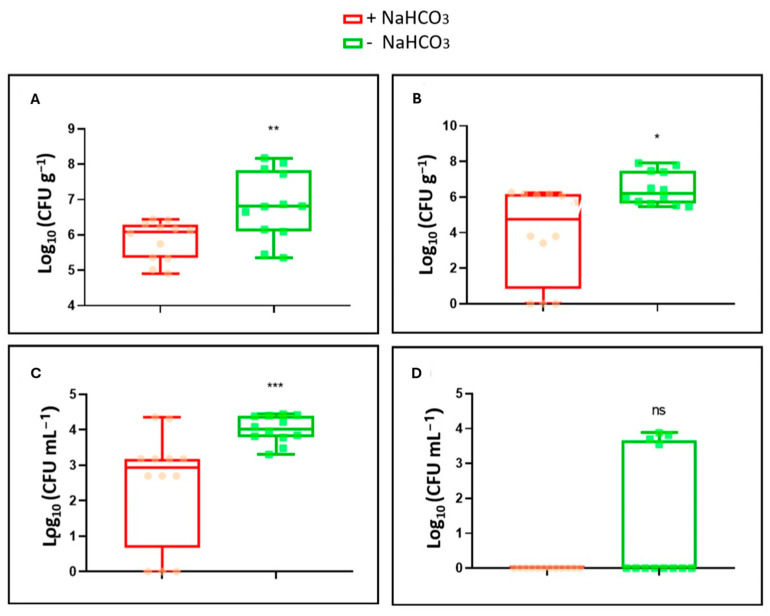
Bacterial colonization of roots and the nutrient solution of tomato plants grown in a hydroponic system. Samples of root and nutrient solution were taken in two periods, one at 26 days and the other at 37 days after inoculation. CFU g^−1^ and CFU mL^−1^ numbers were estimated. (**A**) Root colonization of the WCS417 strain after 26 days in the presence and absence of NaHCO_3_. (**B**) Root colonization of the WCS417 strain after 37 days in the presence and absence of NaHCO_3_. (**C**) Growth of the WCS417 strain after 26 days in nutrient solution in the presence and absence of NaHCO_3_. (**D**) Growth of the WCS417 strain after 37 days in nutrient solution in the presence and absence of NaHCO_3_. * *p* < 0.05, ** *p* < 0.01 and *** *p* < 0.001 indicate significant differences between control and inoculated treatments, ns (no significant differences). Values are the means ± S.E. of 12 replicates.

**Figure 4 plants-14-00264-f004:**
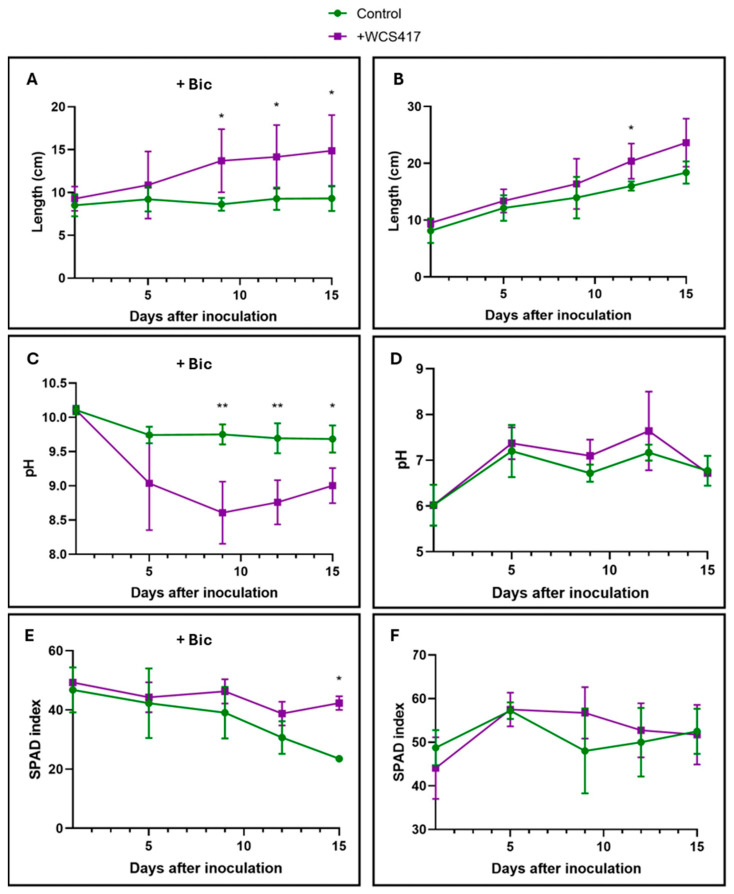
Effect of foliar inoculation with the WCS417 strain on shoot growth, acidification, and SPAD of tomato plants grown in a hydroponic system. After spraying the aerial part of tomato plants with 10^7^ CFU mL^−1^ of strain WCS417, plants were transferred to individual containers. Subsequently, growth and pH evolution over time as well as chlorophyll levels were monitored. (**A**) Growth of tomato plants over 15 days in the presence of 40 mM NaHCO_3_. (**B**) Growth of tomato plants over 15 days in the absence of NaHCO_3_. (**C**) pH evolution in nutrient solution over 15 days of tomato plants grown in the presence of 40 mM NaHCO_3_. (**D**) Evolution of pH in nutrient solution of tomato plants grown in the absence of NaHCO_3_ over 15 days. (**E**) Evolution of chlorophyll levels of tomato plants grown in the presence of 40 mM NaHCO_3_ over 15 days; (**F**) Evolution of chlorophyll levels of tomato plants grown in the absence of NaHCO_3_ over 15 days. * *p* < 0.05, and ** *p* < 0.01 indicate significant differences between control and inoculated treatments. Values are the means ± S.E. of 12 replicates.

**Figure 5 plants-14-00264-f005:**
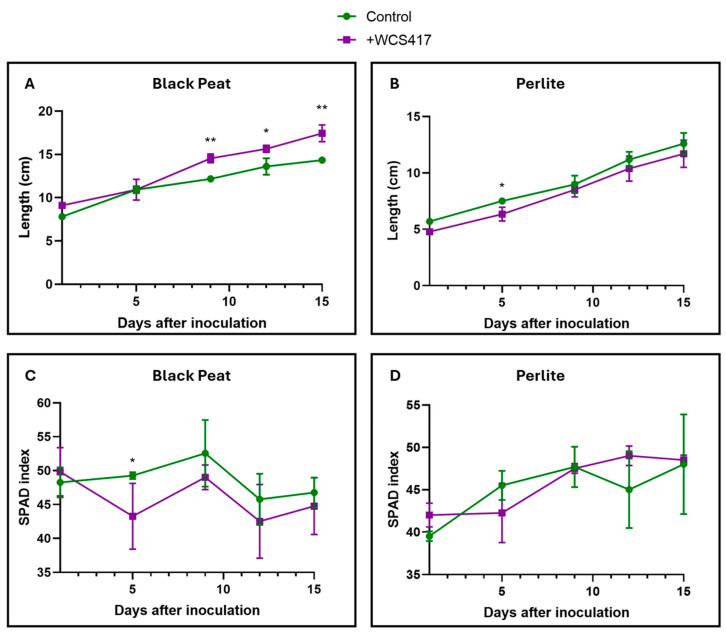
Effect of the inoculation of the WCS417 strain by the irrigation of tomato plants grown in the substrate. After the inoculation of tomato plants with 100 mL of a 10^7^ CFU mL^−1^ bacterial suspension, SPAD and height data were collected for a period of 15 days. (**A**) Height of tomato plants grown in black peat. (**B**) Height of tomato plants grown in perlite. (**C**) Chlorophyll levels of tomato plants grown in black peat. (**D**) Chlorophyll levels of tomato plants grown in perlite. * *p* < 0.05, and ** *p* < 0.01 indicate significant differences between control and inoculated treatments. Values are the means ± S.E. of 12 replicates.

## Data Availability

The original contributions presented in the study are included in the article.: All the data included in this article are publicly available. Further inquiries can be directed to the corresponding author.

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
