# Peer review of "Pseudomonas simiae* WCS417 Strain Enhances Tomato (*Solanum lycopersicum* L.) Plant Growth Under Alkaline Conditions"

_plants, 2025, doi:10.3390/plants14020264_

Round 1

Reviewer 1 Report

Comments and Suggestions for Authors

In this manuscript, the authors performed an interesting study on the beneficial effects of the colonization of roots by growth-promoting rhizobacteria( PGPR) in particular of P. simiae WCS417 strain on the growth of tomato plants subjected to Fe nutrition stress induced by the presence of bicarbonate.

The experimentation was well performed and the methods used are in accordance with analytical criteria.

Data obtained by the authors tend to favor the use of PGPRs that improve fertility components while also stimulating natural soil microflora, reduce phytopathologies, and improve plant healthiness.

The work is original and can be accepted if the authors respond to the following reviewer's comments:

i)                    Line 73 “Regarding the increase in nutrient supply through solubilization, two main

 types of bacterial activities need to be considered.”

 immediately indicate which ones.

Line 73, the following sentence: “On one hand, those that facilitate phosphate

(and other minerals) solubilization since, as a consequence of fertilization, large amounts of

 phosphate accumulate in the soil, but only a small amount of it is available to plants.”

Is not well connected with the previous and distracts the reader.

Rewrite the information provided in a more rational and understandable form.

ii)                  Line 105 “ Also, concerning the stimulation of phytohormone production, as

mentioned in previous chapters”

which chapters?

iii)                Line 55 The authors refer to the nutritional benefits of plants in symbiosis with arbuscular mycorrhizas

I invite the authors to include the bibliographical reference : “Infuence of tomato plant mycorrhization on nitrogen metabolism, growth and fructifcation on P-limited soil. J Plant Growth Regul 38:1183–1195. https://doi.org/10.1007/s00344-019-09923-y

for three kinds of reasons:

a)      Deals with mineral nutrition inherent in the authors' work;

b)     Deals with root capture of phosphorus from soil, well covered in the authors' manuscript

c)      Employs the same biological material as the authors i.e. tomato seedlings

iv)                Although the authors have experimented in hydroponic solution, they cannot disregard the presence of humic and fulvic acids in the soil. A brief comment in the introduction is necessary, also for the consideration that the work is imprinted on calcareous substrates and it would be good to remember:

a)      Humic acids, have chelating properties, bind to nutrients in the soil and facilitate their uptake

b)     High cation exchange capacity (CSC);

c)      Humic acids are soluble in water at alkaline pH. A higher concentration is found in calcareous or neutral soils.

v)                  I urge the authors to add to the action of volatile organic compounds as promoters of Fe chelate reductase activity emitted by Bacillus subtilis GB03 also the action of the well-known ferrobacteria, which as the authors well know are particular schizomycetes capable of promoting oxidative processes at the expense of iron compounds.

In limestone soils where the bicarbonate fraction is in equilibrium with the carbonate fraction, ferrous carbonate is oxidized to insoluble ferric hydroxide with release of CO2 by soil chemoautotrophic ferrobacteria. Such bacteria that the authors are well acquainted with are the architects of the typical red color of the Rio Tinto in Andalusia and which so attracts the interest of scholars.

Authors are invited to add brief commentary on this important aspect of soil microbiology.

vi)                As mentioned above, the experimental results are acceptable and convincing.

vii)              Authors are asked to add the conclusions paragraph to the Discussion paragraph.

Author Response

Reviewer 1

In this manuscript, the authors performed an interesting study on the beneficial effects of the colonization of roots by growth-promoting rhizobacteria (PGPR) in particular of P. simiae WCS417 strain on the growth of tomato plants subjected to Fe nutrition stress induced by the presence of bicarbonate.

The experimentation was well performed and the methods used are in accordance with analytical criteria.

 R: Thank you so much for your thoughtful and kind comment.

Data obtained by the authors tend to favor the use of PGPRs that improve fertility components while also stimulating natural soil microflora, reduce phytopathologies, and improve plant healthiness.

The work is original and can be accepted if the authors respond to the following reviewer's comments:

  R: Thank you so much for your thoughtful and kind comment.

  1. i) Line 73“Regarding the increase in nutrient supply through solubilization, two main

 types of bacterial activities need to be considered.”

 immediately indicate which ones.

R: Thank you very much for your kind suggestion. The manuscript has been updated to include the wo main types of bacterial activities need to be considered, as you wisely suggested.

Line 73, the following sentence: “On one hand, those that facilitate phosphate (and other minerals) solubilization since, as a consequence of fertilization, large amounts of phosphate accumulate in the soil, but only a small amount of it is available to plants.”

Is not well connected with the previous and distracts the reader. Rewrite the information provided in a more rational and understandable form.

 R: The sentence you mentioned has been removed. It was unclear, and following the suggestion of the other reviewer, who advised reducing the introduction, we deemed it appropriate to omit it.

  1. ii)  Line 105“ Also, concerning the stimulation of phytohormone production, as mentioned in previous chapters”

which chapters?

R: I greatly appreciate your comment on this matter. It was an oversight on my part while writing the paper.

iii) Line 55 The authors refer to the nutritional benefits of plants in symbiosis with arbuscular mycorrhizas

I invite the authors to include the bibliographical reference : “Infuence of tomato plant mycorrhization on nitrogen metabolism, growth and fructifcation on P-limited soil. J Plant Growth Regul 38:1183–1195. https://doi.org/10.1007/s00344-019-09923-y

for three kinds of reasons:

  1. a)      Deals with mineral nutrition inherent in the authors' work;
  2. b)     Deals with root capture of phosphorus from soil, well covered in the authors' manuscript
  3. c)      Employs the same biological material as the authors i.e. tomato seedlings

R: The reference you suggested has already been incorporated into the manuscript. Thank you very much for your kind suggestion. The information it contains has enhanced the manuscript. I greatly appreciate the arguments you presented to request the addition of the bibliographic citation.

  1. iv)                Although the authors have experimented in hydroponic solution, they cannot disregard the presence of humic and fulvic acids in the soil. A brief comment in the introduction is necessary, also for the consideration that the work is imprinted on calcareous substrates and it would be good to remember:

  1. a)      Humic acids, have chelating properties, bind to nutrients in the soil and facilitate their uptake
  2. b)     High cation exchange capacity (CSC);
  3. c)      Humic acids are soluble in water at alkaline pH. A higher concentration is found in calcareous or neutral soils.

R: Two bibliographic citations have already been incorporated into the text, which I believe are very interesting and complement the manuscript content perfectly. Thank you very much for your suggestion and for the arguments you presented in support of it.

Delgado, A., Madrid, A., Kassem, S., Andreu, L., & del Carmen del Campillo, M. (2002). Phosphorus fertilizer recovery from calcareous soils amended with humic and fulvic acids. Plant and Soil245, 277-286.

  1. Farid, I., A. El-Ghozoli, M., HH Abbas, M., S. El-Atrony, D., Abbas, H. H., Elsadek, M., ... & Mohamed, I. (2021). Organic materials and their chemically extracted humic and fulvic acids as potential soil amendments for Faba Bean cultivation in soils with varying CaCO3 contents. Horticulturae7(8), 205.

  1. v)   I urge the authors to add to the action of volatile organic compounds as promoters of Fe chelate reductase activity emitted by Bacillus subtilis GB03 also the action of the well-known ferrobacteria, which as the authors well know are particular schizomycetes capable of promoting oxidative processes at the expense of iron compounds.

 In limestone soils where the bicarbonate fraction is in equilibrium with the carbonate fraction, ferrous carbonate is oxidized to insoluble ferric hydroxide with release of CO2 by soil chemoautotrophic ferrobacteria. Such bacteria that the authors are well acquainted with are the architects of the typical red color of the Rio Tinto in Andalusia and which so attracts the interest of scholars.

Authors are invited to add brief commentary on this important aspect of soil microbiology.

R: The paragraph has already been written:

Daliran, T., Halajnia, A., & Lakzian, A. (2022). Thiobacillus bacteria-enhanced iron biofortification of soybean in a calcareous soil enriched with ferrous sulfate, mill scale, and pyrite. Journal of Soil Science and Plant Nutrition, 22(2), 2221-2234.

  1. vi) As mentioned above, the experimental results are acceptable and convincing.

R: Thank you very much for your kind comment. It would be a great opportunity for us to be able to publish this work.

vii)  Authors are asked to add the conclusions paragraph to the Discussion paragraph.

R: As indicated, the conclusions paragraph has already been added to the Discussion paragraph.

Reviewer 2 Report

Comments and Suggestions for Authors

The manuscript, "Pseudomonas simiae WCS417 strain enhances tomato (Solanum lycopersicum L.) plant growth under alkaline conditions," is interesting, but it needs some revisions and more discussion.

Abstract: Please revise the abstract to add more results and conclusions rather than background. The template of Plants shows that abstract needs  (1) Background: Place the question addressed in a broad context and highlight the purpose of the study; (2) Methods: briefly describe the main methods or treatments applied; (3) Results: summarize the articles main findings; (4) Conclusions: indicate the main conclusions or interpretations.

Introduction: Please reduce the contents in the introduction, because some contents are duplicated in the discussion parts, and look like a review paper.

Materials and Methods:

Please add the composition and concentration of the culture solution.

I think the application of 40 mM NaHCO3 is extremely high because tomato plants died in the control treatment. Please add why the authors chose this concentration.

Results:

Figure 1: Please change A and B, and C and D, because the cultivation without NaHCO3 is a control.

A: absence of NaHCO3, B:  +NaHCO3  Please change the same replacement in Figures 4 and 5.

Please show the photos of the plants in the absence of NaHCO3 with and without inoculation.

Discussion:

The results in "3.2 Monitoring the growth of tomato plants and the external pH after foliar inoculation" are very interesting. Please add how the foliar inoculation affected the plant growth promotion. Were the inoculants in the leaves affected, or did they stay in the roots?

Do you think why peat substrate was effective but perlite did not? Please explain or give some hypothesis.

Author Response

Reviewer 2

The manuscript, "Pseudomonas simiae WCS417 strain enhances tomato (Solanum lycopersicum L.) plant growth under alkaline conditions," is interesting, but it needs some revisions and more discussion.

R: Thank you so much for your thoughtful and kind comment.

Abstract: Please revise the abstract to add more results and conclusions rather than background. The template of Plants shows that abstract needs  (1) Background: Place the question addressed in a broad context and highlight the purpose of the study; (2) Methods: briefly describe the main methods or treatments applied; (3) Results: summarize the article’s main findings; (4) Conclusions: indicate the main conclusions or interpretations.

R: I greatly appreciate your feedback. I have structured the abstract as you suggested, and this way, it is much better and clearer.

Introduction: Please reduce the contents in the introduction, because some contents are duplicated in the discussion parts, and look like a review paper.

R: This has already been addressed. The introduction has been significantly condensed to avoid overlaps with the discussion, focusing solely on aspects related to the bacterium's ability to promote growth and the role of bicarbonate in inducing iron (Fe) deficiency.

Materials and Methods:

Please add the composition and concentration of the culture solution.

R: Thank you very much for your kind reminder. I have added the composition of the nutrient solution as you suggested.

I think the application of 40 mM NaHCO3 is extremely high because tomato plants died in the control treatment. Please add why the authors chose this concentration.

R: Our research group published results in 2007 on the effect of bicarbonate on ferric reductase enzyme activity and the expression of certain genes related to iron (Fe) deficiency response mechanisms. In that study, a concentration of 30 mM bicarbonate was used. The experiments were conducted in hydroponic culture, and an inhibitory effect on enzymatic activity and gene expression was observed. However, no negative effects on plant growth were detected.

This time, we decided to slightly increase the bicarbonate concentration, primarily because we wanted to emphasize its negative impact on the growth and development of tomato plants and determine whether bacterial inoculation could counteract this effect. We needed to create a challenging growth environment to better appreciate any positive effects attributable to the presence of the bacterium in the medium.

Differences in development, growth, chlorosis levels, or pH between control and inoculated plants can be very subtle, so we opted for a slightly higher dose than in previous studies to ensure the potential differences between treatments would be more pronounced. Additionally, different cultivation systems (hydroponic and potted solid substrate) were used in this study, and we wanted a dose that would standardize the results across both media.

Results:

Figure 1: Please change A and B, and C and D, because the cultivation without NaHCOis a control.

A: absence of NaHCO3, B:  +NaHCO3  Please change the same replacement in Figures 4 and 5.

R: We kindly request your permission to maintain the current order as presented in the manuscript. From our perspective, we believe this is neither a limiting factor nor something that would hinder the reader’s interpretation of the results. Thanks to your comment, we were able to detect a small error in Figure 5, which has now been corrected. This figure presents results related to the role of the bacterium in plant height and SPAD index in plants grown with two different solid substrates. We have made the necessary adjustments in the figure.

Please show the photos of the plants in the absence of NaHCO3 with and without inoculation.

R: Thank you very much for your kind suggestion. A new photo has been incorporated into the manuscript to replace the previous one. Unfortunately, I do not have photographs of all four treatments you proposed. However, I believe this new image is much more comprehensive and descriptive than the previous one. The new image illustrates a comparison between the control plant, the inoculated plant in the presence of bicarbonate, and the plant in the presence of bicarbonate alone. Thank you for helping to significantly improve the paper.

Discussion:

The results in "3.2 Monitoring the growth of tomato plants and the external pH after foliar inoculation" are very interesting. Please add how the foliar inoculation affected the plant growth promotion. Were the inoculants in the leaves affected, or did they stay in the roots?

R: These are preliminary results. The group's future research efforts will focus on elucidating the bacterium's mode of action. We are working with the hypothesis that foliar application triggers a series of signaling mechanisms that travel from the leaf to the root, activating responses to nutrient deficiency. This, in turn, promotes growth, medium acidification, and increases SPAD values in inoculated plants. Previous studies by the group have shown that hormones such as ethylene or nitric oxide are involved in these signaling systems. A clarifying paragraph has been added to the discussion.

Do you think why peat substrate was effective but perlite did not? Please explain or give some hypothesis.

R: Peat and perlite are distinct solid substrates with different textures and compositions. In the preliminary trials shaping this paper, a viability study of the bacterium in both substrates was not conducted. In our view, irrigation-based inoculation was more successful in peat than in perlite because the bacterium had a greater capacity to remain in the medium. The inoculum may have been washed away with subsequent waterings in perlite but not in peat. For future experiments, we plan to conduct viability analyses in these and other substrates (cultivated soils) to confirm whether the bacterium is suitable for use. A clarifying paragraph has been added to the discussion.

Round 2

Reviewer 2 Report

Comments and Suggestions for Authors

The manuscript has been well revised and this version can be accepted for Plants.